# Chymase as a Possible Therapeutic Target for Amelioration of Non-Alcoholic Steatohepatitis

**DOI:** 10.3390/ijms21207543

**Published:** 2020-10-13

**Authors:** Shinji Takai, Denan Jin

**Affiliations:** Department of Innovative Medicine, Graduate School of Medicine, Osaka Medical College, 2-7 Daigaku-machi, Takatsuki, Osaka 569-8686, Japan; pha012@osaka-med.ac.jp

**Keywords:** non-alcoholic steatohepatitis, inflammation, fibrosis, chymase, inhibitor

## Abstract

The development and progression of non-alcoholic steatohepatitis (NASH) are linked to oxidative stress, inflammation, and fibrosis of the liver. Chymase, a chymotrypsin-like enzyme produced in mast cells, has various enzymatic actions. These actions include activation of angiotensin II, matrix metalloproteinase (MMP)-9, and transforming growth factor (TGF)-β, which are associated with oxidative stress, inflammation, and fibrosis, respectively. Augmentation of chymase activity in the liver has been reported in various NASH models. Generation of hepatic angiotensin II and related oxidative stress is upregulated in NASH but attenuated by treatment with a chymase inhibitor. Additionally, increases in MMP-9 and accumulation of inflammatory cells are observed in NASH but are decreased by chymase inhibitor administration. TGF-β and collagen I upregulation in NASH is also attenuated by chymase inhibition. These results in experimental NASH models demonstrate that a chymase inhibitor can effectively ameliorate NASH via the reduction of oxidative stress, inflammation, and fibrosis. Thus, chymase may be a therapeutic target for amelioration of NASH.

## 1. Introduction

Non-alcoholic fatty liver disease (NAFLD) is characterized by over-accumulation of fat in the liver (≥5%) that is not related to alcohol consumption and viral infection. Approximately 10% of NAFLD patients progress to non-alcoholic steatohepatitis (NASH), which is characterized by inflammation and fibrosis, progressing into cirrhosis and hepatocellular carcinoma [1,2]. Development of NAFLD progression and NASH are associated with metabolic syndrome resulting from obesity, insulin resistance, hyperlipidemia, and hypertension [3,4,5]. NAFLD is the most common liver disease and typically presents as simple hepatic steatosis, and NASH is characterized by severe steatosis, lobular inflammation, liver fibrosis [6,7]. The progression of NAFLD to NASH is thought to be caused by a “multiple-hit” process, with hepatic steatosis as the “first hit” and subsequent hits such as oxidative stress and inflammation [8]. Although clinical studies have investigated therapeutic treatment of NASH with attenuating agents of diabetes, hyperlipidemia and hypertension, no commonly accepted therapeutic agents have been established [3,4,5].

Chymase is a chymotrypsin-like enzyme that is expressed in the secretory granules of mast cells [9]. Chymase is synthesized as an inactive prochymase, and dipeptidyl peptidase I (DPPI) is necessary for chymase activation within secretory granules [10,11]. This activity is regulated at pH 5.5, which is the optimal pH of DPPI in secretory granules, and DPPI can activate prochymase to chymase [10,11]. However, the optimal pH of chymase ranges between 7 and 9, and it does not show enzymatic activity at pH 5.5; thus, chymase is inactive within mast cell granules [11]. Chymase exhibits enzymatic activity immediately following its release from mast cell granules because interstitial fluid has a pH of 7.4. An enzymatic function of chymase was originally discovered in canine vascular extracts as an angiotensin II-producing enzyme other than angiotensin converting enzyme (ACE) [12]. Subsequently, it was shown that heart and blood vessel extracts also contain an angiotensin II-forming chymase in several species including humans [13,14,15,16]. Angiotensin II-forming ability is an enzymatic function of chymase, and activation of matrix metalloproteinase (MMP)-9 and transforming growth factor (TGF)-β from each precursor is also demonstrated [9]. Angiotensin II activates nicotinamide adenine dinucleotide phosphate (NADPH) oxidase via stimulation of its receptor and induces oxidative stress other than the regulation of blood pressure. MMP-9 and TGF-β are closely related to inflammation and fibrosis, respectively. Therefore, augmentation of chymase enzymatic function results in upregulation of oxidative stress, inflammation, and fibrosis.

Chymase activity is significantly increased in livers of cirrhosis patients and a significant correlation between chymase level and fibrotic score has been shown [17]. Although chymase activity has not been reported in NASH patients, it is increased in the liver from various NASH models [18,19,20,21]. In contrast, chymase inhibitors dramatically attenuate angiotensin II, MMP-9, and TGF-β reduction observed in NASH, subsequently improving steatosis, inflammation, and fibrosis [18,19,20]. In this review, we propose the possibility of chymase as a useful target for amelioration of NASH.

## 2. Enzymatic Functions of Chymase on NAFLD/NASH

Chymase is a chymotrypsin-like serine protease that cleaves the C-terminal side of proteins after aromatic amino acids such as Phe, Tyr, and Trp [9]. Chymase can cleave non-bioactive peptide angiotensin I to form its bioactive peptide angiotensin II. Chymase also cleaves MMP-9 and TGF-β precursors to their active forms [9]. In experimental NASH models, chymase activity is increased along with MMP-9 and TGF-β levels. Furthermore, chymase cleaves the inactive membrane-bound form of stem cell factor (SCF) to form SCF, which induces the formation of mature mast cells from immature mast cells via c-kit receptor stimulation [22]. This enzymatic function of chymase may promote increased chymase-containing mast cell number in NASH. In fact, chymase inhibitors have been observed to reduce chymase-positive mast cell numbers in various experimental NASH models [18,19,20]. Enzymatic functions of chymase and their actions are summarized in Table 1.

Chymase plays an important role in the formation of tissue angiotensin II [9]. However, ACE is better known as an angiotensin II-forming enzyme than chymase. ACE inhibition attenuates angiotensin II functions, such as vascular constriction, resulting in reduced blood pressure. Previously, we studied a two-kidney, one clip (2K1C) hypertensive model of hamster [23]. The blood pressure in the 2K1C hamster increased significantly 2 weeks after clipping (acute stage), and was sustained at the high level until 32 weeks after clipping (chronic stage). Plasma renin activity increased significantly during the acute stage, but returned to the normal level at the chronic stage. In the chronic stage, the vascular ACE activity, but not the chymase activity, increased significantly, and an ACE inhibitor and an angiotensin II receptor blocker (ARB) showed equipotently hypotensive effect at the acute and chronic stages [23]. ACE inhibitors and ARBs are known not only to reduce blood pressure but also to increase plasma renin activity. Conversely, chymase inhibitors did not reduce blood pressure in several hypertensive models [20,24]. Because chymase has no enzymatic activity within mast cell granules [11], and only becomes active immediately following its release into interstitial tissues after stimulation. However, the enzymatic functions may be unable to be observed in plasma, because endogenous chymase inhibitors such as α-antitrypsin and α-antichymotrypsin have been found in plasma, and they inhibit chymase activity immediately, in a few seconds [25]. Therefore, chymase inhibitors could attenuate angiotensin II formation at the tissue level, where it occurs with inflammation but does not interfere with blood pressure homeostasis. Angiotensin II induces hepatic steatosis and inflammation by increasing reactive oxygen species (ROS) following stimulation of type 1 angiotensin II receptors in animal NASH models [26,27]. Angiotensin II also promotes hepatic fibrosis via induction of α-smooth muscle actin (SMA) in hepatic stellate cells (HSC) [28].

MMP-9 induces neutrophil and macrophage infiltration via cleavage of intercellular matrixes such as vitronectin and fibronectin, subsequently augmenting inflammation. In the rat indomethacin-induced small intestinal injury model, a chymase inhibitor TY-51469 reduced MMP-9 activity in the small intestine, preventing small intestinal damage [29]. Myeloperoxidase activity, which indicates neutrophil accumulation, was significantly increased in the small intestine after indomethacin administration, but its activity was significantly attenuated by treatment with a chymase inhibitor. Neutrophils express proMMP-9, and chymase inhibition may result in attenuating not only MMP-9 activity but also proMMP-9 expression levels. In NASH patients, MMP-9 gene expression was significantly increased in the liver compared to normal controls, while MMP-2 was increased in patients with chronic hepatitis B and hepatitis C [30]. Thus, MMP-9 activation derived from chymase may be associated with inflammation in NASH.

TGF-β formation induces fibrosis via increased collagen expression. Bleomycin is known to cause lung fibrosis in humans [31]. TGF-β may play an important role in the development of bleomycin-induced pulmonary fibrosis in animal models [32]. For example, administration of anti-TGF-β antibodies could reduce bleomycin-induced pulmonary fibrosis via reduction of collagen levels [33]. In hamsters, significant increases in chymase activity and fibrotic areas in pulmonary tissues after bleomycin were significantly reduced by a chymase inhibitor NK3201 [34]. In bleomycin-induced pulmonary fibrosis in mice, a chymase inhibitor SUN-C8077 significantly attenuated the increase in chymase and pulmonary hydoxyproline levels [35]. In silica-induced pulmonary fibrosis in mice, a chymase inhibitor led to a significant reduction in chymase activity and pulmonary fibrosis as well as TGF-β level [36]. In transgenic mice, hepatic overexpression of TGF-β produced severe hepatic fibrosis [37]. Chymase-dependent TGF-β activation may also be deeply involved in the development and progression of NASH. These enzymatic functions of chymase may be involved in steatosis, inflammation, and fibrosis, which are observed in the livers of NASH patients and animal models (Figure 1).

The mast cell stabilizer tranilast prevents mast cell activation, blocking the release of chymase and thereby preventing the development of hepatic fibrosis in a rat metabolic syndrome-induced NASH model [38]. Chymase promotes the proliferation of mast cells via SCF activation by its enzymatic function [22]. In NASH experimental models, chymase inhibitor reduced increases in mast cell number in NASH, reducing chymase activity following both direct inhibition by chymase inhibitor and indirect reduction of chymase expression in mast cells [19,20].

## 3. Experimental Evidences on Chymase Inhibitors in NAFLD/NASH

### 3.1. Effects of Chymase Inhibitors on Hepatic Steatosis

Angiotensin II may influence hepatic steatosis via ROS generation, such as superoxide and hydrogen peroxide, through the activation of NADPH oxidase in HSC [39]. An ARB significantly attenuated hepatic steatosis in a rat metabolic syndrome-induced NASH model [40]. In a mouse MCD diet-induced NASH model, a significant attenuation of steatosis was observed in angiotensin II receptor-deficient mice [27]. An NADPH oxidase inhibitor significantly decreased ROS production in mouse HSC, and an ARB slowed the development of hepatic steatosis via attenuation of ROS production in rat NASH models [26,41]. In experimental NASH models, angiotensin II upregulates sterol regulatory element-binding protein (SREBP)-1c and fatty acid synthase (FAS) gene expression, which are important factors in lipogenesis regulation, following augmentation of ROS generation [42,43]. Conversely, an ARB attenuated hepatic steatosis along with downregulation of SREBP-1c and FAS gene expression via attenuation of ROS generation in a mouse NASH model [44]. In a hamster MCD diet-induced NASH model, significant attenuation of SREBP-1c and FAS gene expression was observed along with reduced hepatic angiotensin II concentration by treatment with chymase inhibitor [19]. Especially, SREBP-1 is a key membrane-bound transcription factor via promotion of hepatic lipid synthesis by insulin. High plasma glucose and insulin levels after eating carbohydrates induce rapid synthesis of long-chain fatty acids (LCFA) in hepatocytes. Elevated plasma insulin promotes glucose transporter-4 (GLUT-4)-mediated plasma glucose uptake adipocytes. Reduction of GLUT-4 promotes postprandial hyperglycaemia and insulin release, but an ARB losartan improves the hyperglycemia and insulin resistance via an increase in GLUT-4 in obese Zucker rats [45]. Increases in GLUT-4 and tumor necrosis factor (TNF)-α drives hepatic LCFA synthase from acyl CoA, and the conversion of LCFA to triglyceride by glycerol phosphate acyl transferase within hepatocytes will increase hepatocyte triglyceride synthase and accumulation [46]. Therefore, development of hepatic steatosis may be intimately involved in the increase in angiotensin II produced by chymase.

### 3.2. Effects of Chymase Inhibitors on Hepatic Inflammation

Acute liver failure (ALF) is one of the inflammatory disease of the liver, but it was not a common chronic inflammatory hepatic diseases such as NASH. Although the inflammatory mechanism of ALF is thought to be different from that of NASH, we present it as an inflammatory mechanism in the liver. Human ALF is initiated by activation of T cells and macrophages, causing the release of proinflammatory cytokines such as TNF-α, leading to massive death of liver parenchyma cells [47]. The combined administration of lipopolysaccharide and D-galactosamine (LPS/Galn) has become an established method of ALF induction in experimental models. This type of ALF model is a useful tool for the evaluation of drugs that attenuate liver inflammation. Lipopolysaccharides are a major component of endotoxin in Gram-negative bacteria and are frequently implicated in the cause of liver failure [48]; D-galactosamine is an amino sugar that is selectively metabolized by hepatocytes and induces transcriptional repression in the liver [49]; D-galactosamine potentiates the toxic effects of lipopolysaccharides and causes necrosis and apoptosis [50]. Stimulation of hepatic macrophages by lipopolysaccharide causes secretion of pro-inflammatory cytokines, such as TNF-α, which causes liver failure in rodents [51]. In a mouse model of ALF, MMP inhibitors reduced plasma alanine aminotransferase and MMP-9 levels and inhibited the accumulation of macrophages and neutrophils in the liver [52]. These phenomena have also been observed in the LPS/GalN-induced ALF model in MMP-9-deficient mice [53]. In a hamster ALF model, hepatic chymase activity was significantly increased after the administration of lipopolysaccharide and D-galactosamine [54]. Significant increases in hepatic MMP-9 activity and TNF-α concentration were also observed, but these increases were attenuated by treatment with a chymase inhibitor, TY-51469 [54]. Significant increases in plasma aspartate aminotransferase and alanine aminotransferase activities were significantly attenuated by chymase inhibition [54]. Chymase inhibition may provide a therapeutic strategy for attenuating the symptoms of hepatic inflammation.

Hepatic MMP-9 levels are significantly increased along with chymase activity in NASH [6,7,8]. Augmentation of ROS generation induced by angiotensin II promotes MMP-9 gene expression in neutrophils and macrophages [55,56]. Chymase can also directly promote the activation of proMMP-9 to MMP-9 in vitro [57]. Therefore, chymase induces MMP-9 activity not only via indirect upregulation of MMP-9 gene expression but also via direct progression of MMP-9 activation from proMMP-9. MMP-9 cleaves extracellular matrix constituents, such as vitronectin and fibronectin, leading to the disintegration of integrity and inducing macrophage and neutrophil infiltration [58]. Neutrophil accumulation was observed in the skin where purified human chymase or mouse chymase was injected [59,60]. These reports clearly suggest that chymase directly provides a potent stimulus for neutrophil recruitment. Furthermore, in a hamster metabolic syndrome-induced NASH model, a significant increase in myeloperoxidase level from macrophages and neutrophils was observed in the liver, but this increase was ameliorated by treatment with chymase inhibitor [20]. Therefore, augmented chymase may induce inflammatory cell accumulation via increased angiotensin II generation and MMP-9 concentrations in the liver.

### 3.3. Effects of Chymase Inhibitors on Hepatic Fibrosis

Mast cells are present in the liver; their numbers have been reported to increase in chronic liver diseases associated with fibrosis in human and experimental models [17,61]. Chymase has also been closely associated with the progression of tissue fibrosis because it promotes the formation of TGF-β from the non-bioactive precursor TGF-β, and TGF-β strongly induces fibroblast growth [62]. TGF-β may play a central role in the progression of fibrosis in NASH patients via activated HSC [63]. Inhibition of TGF-β function via gene expression and signaling resulted in improved hepatic fibrosis in experimental models [64,65]. In a rat metabolic syndrome-induced NASH model, attenuation of chymase activity by chymase inhibitor reduced TGF-β expression and fibrotic areas in the liver [20]. Thus, chymase-dependent TGF-β augmentation may be closely involved in hepatic fibrosis in NASH.

Angiotensin II is also involved in the induction of hepatic fibrosis. In metabolic syndrome-induced NASH model, an ARB could prevent fibrosis [40]. Angiotensin II induces HSC proliferation and TGF-β gene expression in fibroblasts [66,67]. In angiotensin II receptor-deficient mice, TGF-β levels, collagen accumulation, and fibrotic lesions by bile duct ligation were attenuated [68]. HSC are recognized as the main collagen-producing cells in the liver, and augmentation of α-SMA expression in HSC progresses collagen I. Angiotensin II blockage results in hepatic fibrosis attenuation along with α-SMA reduction [28]. Both chymase and angiotensin II-forming activities were significantly augmented in fibrotic regions of livers from patients with cirrhosis, and significant correlations among chymase, angiotensin II-forming activity, and hepatic fibrosis were observed [17]. Both gene expressions of chymase and angiotensin II receptor were significantly increased in human cirrhotic livers, and angiotensin II blockade benefited patients with chronic hepatitis C after liver transplantation by reducing the probability of cirrhosis [69,70]. In a hamster tetrachloride-induced hepatic cirrhosis model, not only significant increases in chymase and angiotensin II-forming activity but also significant correlations among chymase, angiotensin II-forming activity, and hepatic fibrosis were observed, and chymase inhibitor could prevent hepatic cirrhosis [17]. Taken together, chymase may progress hepatic fibrosis via TGF-β activation by chymase and TGF-β gene expression augmentation via increased angiotensin II generation.

### 3.4. Effects of Chymase Inhibitors on Experimental NASH Models

The MCD diet-induced NASH model has been used to evaluate preventive effects of therapeutic agents for NASH. In a hamster MCD diet-induced NASH model, increased plasma total bilirubin, triglyceride, and hyaluronic acid levels were significantly reduced by administration of a chymase inhibitor TY-51469 at the same time as MCD diet loading [18]. Substantial hepatic steatosis and fibrosis augmentation, which is observed in NASH, were significantly prevented by chymase inhibition [18]. In addition to chymase activity, both angiotensin II-forming and MMP-9 activities were significantly increased in the liver of the NASH model. In contrast, the chymase inhibitor blocked both angiotensin II-forming and MMP-9 activities in addition to chymase activity, and significantly attenuated the hepatic steatosis and fibrosis [18]. These findings demonstrate that augmentation of angiotensin II-forming and MMP-9 activities is closely associated with increased chymase activity in NASH. In this model, chymase inhibition was shown to be useful for preventing NASH. Using this same NASH model, not only the preventive effect of chymase inhibitor, but also an improving effect was seen [19]. A chymase inhibitor TY-51469 was started 12 weeks after the MCD diet load when NASH was already formed and was continued concurrently with the MCD diet for an additional 12 weeks [19]. At 12 weeks after the start of the study prior to the administration of the chymase inhibitor, marked hepatic steatosis and fibrosis were observed, and both chymase activity and angiotensin II concentration were also increased. In this study, both hepatic steatosis and fibrosis were significantly reduced by treatment with the chymase inhibitor, and there were no significant differences compared with normal diet [19]. Furthermore, increased collagen I and collagen III mRNA levels in addition to chymase activity and angiotensin II concentration were reduced by the chymase inhibition. Therefore, chymase inhibition has been shown to improve NASH in addition to preventing it in this hamster MCD diet-induced NASH model.

As a characteristic NASH model, the MCD diet-induced NASH model has been used in experimental studies. This model is useful for evaluating hepatic steatosis and fibrosis, but it does not recognize symptoms of metabolic syndrome such as obesity, hyperlipidemia, diabetes, and hypertension. Recently, we published a hamster metabolic syndrome-induced NASH model. Model hamsters are fed a high-fat and high-cholesterol diet and exhibit obesity and symptoms of hyperlipidemia, diabetes, and hypertension [21]. In this model, angiotensin II-forming activity was significantly increased in NASH along with increased chymase activity and upregulation of MMP-9 and TGF-β levels [21]. Another experimental model employed is the rat metabolic syndrome-induced NASH model, which is developed using stroke-prone spontaneously hypertensive 5 (SHRSP5)/Dmcr rats fed high-fat and high-cholesterol (HFC) diet. Model rats exhibit symptoms of hypertension and hyperlipidemia. Using this model, preventive and ameliorative effects of a chymase inhibitor TY-51469 on NASH were evaluated [20]. SHRSP5/Dmcr rats were fed HFC diet for 8 weeks, and concurrently administered either placebo or TY-51469. In this model, a significant reduction in plasma triglyceride level was observed in the TY-51469-treated group compared with the placebo-treated group. The chymase level was significantly increased along with MMP-9 and TGF-β levels in NASH, but all increased levels were significantly reduced by the chymase inhibitor. Hepatic steatosis and fibrosis observed in NASH were significantly attenuated by administering chymase inhibitor [20]. Furthermore, to evaluate the effect of TY-51469 on the survival rate, it was administered concurrently with HFC diet (pretreated group) and 8 weeks after the start of HFC diet at which point NASH had developed (posttreated group) [20]. A survival rate of the placebo-treated group was 0% at 14 weeks. In comparison, the rates of TY-51469-pretreated and TY-51469-posttreated groups were 100% and 50% at 14 weeks, respectively [20]. These findings strongly demonstrated that chymase may be a useful target for metabolic syndrome-induced NASH.

The changes of chymase-related factors and effects of chymase inhibitors in experimental NASH models are summarized in Table 2.

## 4. Role of Chymase on Metabolic Syndrome Complications

Chymase is also associated with complications of the metabolic syndrome other than NASH [71]. In rat models of hypertension, chymase does not appear to regulate blood pressure, but may be intimately involved in promoting vascular damage [24,72]. In stroke-prone spontaneously hypertensive rats (SHR-SPs), chymase inhibitors significantly improved vascular dysfunction and significantly prolonged cumulative survival compared to placebo [24]. Lifespan extension by chymase inhibition may be associated with the prevention of vascular fragility in SHR-SPs. Hypertension is one of the major determinants of vessel wall structure and composition. Increased shear stress on the vascular endothelium may increase MMP-9 levels and increase ROS [73]. Chymase inhibition attenuated the increase in MMP-9 activity and oxidative stress in SHR-SPs [24]. In human essential hypertension, plasma MMP-9 levels were higher than in normotensive patients [74,75]. In hypertensive rats, plasma MMP-9 levels were higher than in normotensive rats, and an MMP inhibitor significantly attenuated vascular remodeling without lowering blood pressure in SHR-SPs [76,77]. Attenuation of MMP-9 by chymase inhibition may contribute to the attenuation of vascular endothelial dysfunction in hypertension as well as to the prevention of vascular fragility.

The number of activated mast cells containing chymase was increased in human atherosclerotic lesions [78]. Atherosclerosis is closely related to hypercholesterolemia as well as hypertension. Chymase activity was increased in arteries in patients with hypercholesterolemia, and serum cholesterol level was significantly correlated with vascular chymase activity [79]. A significant correlation between plasma low-density lipoprotein cholesterol and vascular chymase activity was also observed in high-cholesterol-fed hamsters, but a chymase inhibitor reduced atherosclerotic lesions [80]. In apolipoprotein E-deficient mice, the progression of atherosclerotic plaques was prevented by chymase inhibition [81]. In a high-cholesterol diet-induced atherosclerosis monkey model, both chymase and angiotensin II concentration was significantly increased in atherosclerotic lesions, but ARBs significantly improved atherosclerotic plaque without affecting blood cholesterol levels [16,82]. The chymase-dependent angiotensin II formation in vascular tissue may be related to the development of atherosclerosis. On the other hand, the involvement of chymase in the development of atherosclerosis may also include the modification of high-density lipoprotein (HDL) by chymase. HDL cholesterol plays a crucial role in cholesterol efflux from macrophages, and the modification of HDL cholesterol by chymase may lead to decrease cholesterol efflux [83,84]. Thus, in addition to the angiotensin II formation in vascular tissue, chymase may be involved in atherosclerosis via HDL modification in the blood.

Chymase may also be involved in the progression of diabetic complications. Chymase-dependent angiotensin II may play a crucial role in podocyte injury. A significant increase in glomerular chymase level was observed in patients with diabetic nephropathy [85]. High glucose increases angiotensin II formation in cultured mouse podocytes, and its increase can be inhibited by a chymase inhibitor but not by an ACE inhibitor [86]. A chymase inhibitor Suc-Val-Pro-Phe-^P^(OPh)_2_ reduced urinary albumin excretion and deposition of extracellular matrix components in the kidney of type 1 diabetic rats [87]. In a hamster type 1 diabetic model, myocardial fibrosis via the attenuation of angiotensin II concentration and oxidative stress in heart was attenuated by treatment with chymase inhibitors, TEI-F00806 and TEI-E00548 [88]. In a mouse type 2 diabetic model, a chymase inhibitor Suc-Val-Pro-Phe^p^(OPh)_2_ reduced albuminuria and renal angiotensin II concentration [89]. In a streptozotocin-induced hamster diabetic model, a significant increase in blood glucose level was observed along with increases in chymase and angiotensin II-forming activities in pancreatic islets, and a chymase inhibitor attenuated the increases in blood glucose level [90]. Conversely, a significant increase in blood glucose levels was observed after streptozotocin injection in human chymase transgenic mice compared with normal mice [91].

Therefore, preventive and improving effects of NASH by chymase inhibitors may be indirectly influenced by protective effects of organs in metabolic syndrome complications.

## 5. Summary and Future Prospects

Chymase plays an important role in hepatic inflammation and fibrosis in experimental NASH models via various enzymatic functions, including upregulation of angiotensin II, MMP-9 and TGF-β. Effects of chymase inhibitors have been shown against a variety of diseases in animal experimental models. The enzymatic function of chymase was first found in the heart and blood vessels, and their related diseases have been well studied in animal models. In models of myocardial infarction, chymase inhibition has been shown to improve cardiac function dysfunction and prevent arrhythmias [92,93]. In a phase II clinical trial, effects of a chymase inhibitor fulacimstat on adverse cardiac remodeling after acute myocardial infarction was evaluated [94]. Although safe and well tolerated in patients with left-ventricular dysfunction after acute myocardial infarction, chymase inhibition did not significantly improve the cardiac dysfunction [94]. In this study, most patients were treated with β-blockers and renin-angiotensin-related drugs that have shown efficacy in heart failure, and it might be difficult to show the additional effect. Moreover, chymase activity was significantly increased even 1 day after myocardial infarction in an animal study, but administration of chymase inhibitor was initiated from 6 days to 12 days after myocardial infarction in this clinical study [92,94]. It may have better started administering the administration earlier. On the other hand, a phase II clinical trial for diabetic nephropathy with a chymase inhibitor fulacimstat has been completed, but the results of this trial have not yet been released. Clinical trials for patients with NASH have not yet been conducted and are expected to be conducted in the future.

## Figures and Tables

**Figure 1 ijms-21-07543-f001:**
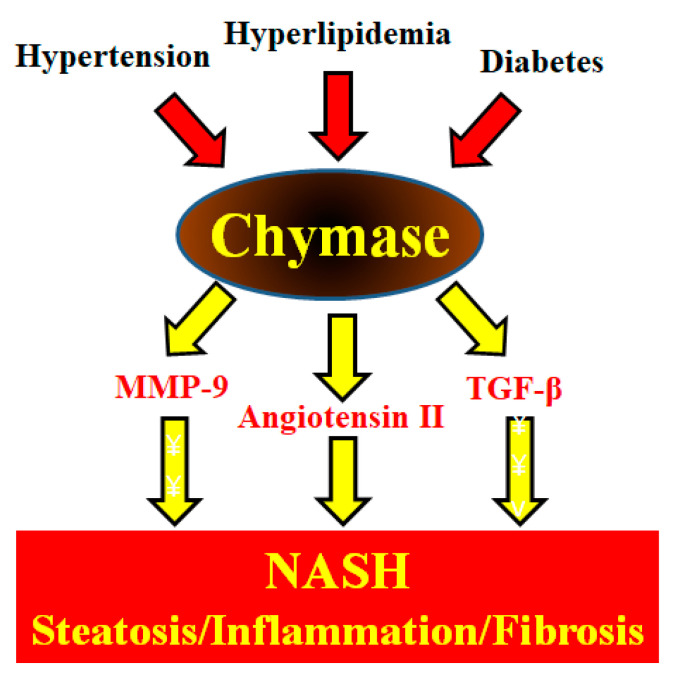
Metabolic syndrome (hypertension, hyperlipidemia, and diabetes) augments hepatic chymase levels. Chymase induces the development and progression of NASH symptoms such as hepatic steatosis, inflammation, and fibrosis via its enzymatic functions.

**Table 1 ijms-21-07543-t001:** Enzymatic function of chymase and their actions.

Enzymatic Functions of Chymase	Actions via Enzymatic Functions of Chymase
Angiotensin II formation	Increase of oxidative stress
Activation of TGF-β	Increase of tissue fibrosis
Activation of MMP-9	Increase of inflammatory cell accumulation
Activation of SCF	Increase of mast cell number

**Table 2 ijms-21-07543-t002:** Summary of experimental NASH models.

	MCD Diet-Induced Hamster NASH Model [18,19]	Metabolic Syndrome- Induced NASH Model [20]
Chymase-	Chymase ⬆	Chymase ⬆
related factors	Mast cells ⬆	Mast cells ⬆
	Angiotensin II ⬆	MMP-9 ⬆
	MMP-9 ⬆	Myeloperoxidase ⬆
	Malondialdehide ⬆	TGF-β ⬆
Effects of	Hapatic steatosis ⬆	Hapatic steatosis ⬆
chymase inhibitor	Hepatic fibrosis ⬆	Hepatic fibrosis ⬆
	Plasma triglyceride ⬆	Hepatic inflammation ⬆
	Plasma hyaluronic acid ⬆	Plasma triglyceride ⬆
	Body Weight ⬌	Plasma cholesterol ⬆
	Liver wight/body weight ⬌	Body Weight ⬌
		Liver wight/body weight ⬌
		Hypertension ⬌

⬆: Increase; ⬌: No change.

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
