# Peer review of "Chymase as a Possible Therapeutic Target for Amelioration of Non-Alcoholic Steatohepatitis"

_ijms, 2020, doi:10.3390/ijms21207543_

Round 1

Reviewer 1 Report

In the review entitled "Chymase as a possible therapeutic target for amelioration of non-alcoholic steatohepatitis", Takai and Jin summerized the pathophysiological role of the enzyme chymase in the setting of non-alcoholic steatohepatitis suggesting its potential role as therapeutic target for the improvement of non-alcoholic steatohepatitis.

Overall, it is a well-written review even if some concepts should be better explained.

Comments:

  • I suggest to add a table with a list of mechanisms in which chymase and its inhibitors are involved. The figure help the readers to understand through which factors chymase induces steatosis, inflammation and fibrosis but a table summarizing the most important studies performed in vitro or in vivo it would be better.
  • Page 1, lines 35-39 --> the link between chymase and oxidative stress should be better explained. It is not clear
  • Page 1, line 41 --> please, consider to change the sentance as suggested: Non-alcoholic Fatty Liver Disease (NAFLD) is characterized by an over-accumulation of fat in the liver (>=5%) that is not related to alcohol consumption and viral infection.
  • Page 4, line 176 -> the mechanism described could be misinterpreted because is the excessive accumulation of fat in the liver that causes oxidative stress through lipotoxicity exacerbating inflammation and promoting fibrogenesis; probably is angiotensin II that is modulated in a microenvironment characterized by a low grade of chronic inflammation that is typical in patients with NAFL and NASH.
  • Page 5, lines 198-199 --> as stated before, I suggest to revise the sentence accordingly
  • Page 5, line 201 --> Acute liver failure (ALF) is not a common chronic inflammatory disease! Acute liver failure (ALF) refers to a rare syndrome characterised by an acute liver injury resulting in encephalopathy (altered mentation) and coagulopathy (...) in individuals without known pre-existing liver disease and (...) (Hepatology 2012;55:965-7). Please, pay attention to the acronym
  • Page 7, line 320 --> "In this study, 'all' or 'most' ..." ?

Author Response

Thank you for your valuable suggestions.

We added a table with enzymatic functions of chymase (Table 1).

Page 2, lines 66-69 (revised manuscript)

Enzymatic functions of chymase and their actions are summarized in Table 1.

We added a table with summary of experimental animal studies (Table 2).

Page 6, lines 250-253(revised manuscript)

The changes of chymase-related factors and effects of chymase inhibitors in experimental NASH models are summarized in Table 2.

Page 1, lines 36-39 (revised manuscript: pages 1-2, lines 46-48)

We added an explanation to the text.

Angiotensin II activates nicotinamide adenine dinucleotide phosphate (NADPH) oxidase via stimulation of its receptor and induces oxidative stress other than the regulation of blood pressure.

Page 1, line 41 (revised manuscript: page 1, lines 24-25)

We changed the sentence as you suggested.

NAFLD is characterized by over-accumulation of fat in the liver (>5%) that is not related to alcohol consumption and viral infection.

Page 5, lines 198-199 (revised manuscript: page 4, lines 160)

We deleted the sentence “Pro-inflammatory cytokines are potent inducers of MMPs [52]”.

Page 5, line 201 (revised manuscript: page 4, line 148-150).

As you suggested, we specified that the inflammatory mechanism of ALF is different from that of NASH.

Acute liver failure (ALF) is one of the inflammatory disease of the liver, but it was not a common chronic inflammatory hepatic diseases such as NASH. Although the inflammatory mechanism of ALF is thought to be different from that of NASH, we present it as an inflammatory mechanism in the liver.

Page 7, line 300 (revised manuscript: page 7, line 308-310)

We corrected the sentence.

In this study, most patients were treated with β-blockers and renin-angiotensin-related drugs that have shown efficacy in heart failure, and it might be difficult to show the additional effect.

Reviewer 2 Report

Takai et al. have reviewed the possibility of chymase as a possible therapeutic target for amelioration of non-alcoholic steatohepatitis. They have thoroughly discussed the literature available on chymase and possible underlying mechanisms through which NASH can be targeted. The authors have discussed different NASH models and implication of using chymase as a target in the process of attenuation of NASH. They have also discussed pathophysiological roles in hepatic fibrosis, inflammation and steatosis.

Even though the authors have discussed chymase as a potential target in detail, this article is not recommended to be published in this esteemed journal. The main reason is the authors have published a similar article (https://pubmed.ncbi.nlm.nih.gov/29515450/)

back in 2018 in the journal "Front pharmacology" titled "Chymase Inhibitor as a Novel Therapeutic Agent for Non-alcoholic Steatohepatitis". This article has a significant overlap with the current manuscript with very little new information added. In my opinion after reading this manuscript and the article published, a new review for this topic is not warranted as of this time. In future, if the significant advancements are published then the review on this matter will help gather all the findings at one place.

Author Response

Thank you for your valuable suggestions.

As you pointed out, this review is similar to it in Front Pharmacology. The reason for this is that the editor saw the review in Front Pharmacology review article and asked us to submit the review about NASH to IJMS.

In accordance with reviewer 3, we have changed the content significantly and added the role of chymase in complications of metabolic syndrome other than NASH. We believe that the content is different from that of Front Pharmacology.

Reviewer 3 Report

Important topic, well selected in the middle of feverish search for new targets in NAFLD/NASH, and comprehensive (full of supporting evidence)

However: 

The review is boring, and reader will loose attention after the first of endless   repetitions on AT, MMP, and TGF pathways.

It is easily solvable, however. I suggest to consider two steps:

  • re-structuring the sections
  • Substantial shortening of all the non-NAFLD subchapters
  • Adding table, summarizing all the evidence in NAFLD/NASH, with columns a)targeted item in rows (steatosis, inflammation, fibrosis, other), b)and in columns: inhibitor generic name / mechanism (ROS, AT, MMP, TGF,...); type of study & primary outcome & number of animals/patients; outcome; comment

Suggested sections in re-structured text:

Section 1 Introduction on NAFLD/NASH

Section 2 Intro on Physiology of Chymase & Theory

Section 3 General review of experimental evidence on inhibitors in NAFLD/NASH, sorted by:

  • effects on steatosis
  • effect on inflammation
  • effect on fibrosis

Section 4 Clinical evidence in NAFLD/NASH (however poor)

Section 5 Clinical evidence from other syndromes (metabolic syndrome, pulmonary, intestinal, renal - as provided in the current version, but with substantial shortening)

Section 6 Summary and future prospects

Author Response

Thank you for your valuable suggestions.

According to your suggestion, I have made the following five sections.

  1. Introduction
  2. Enzymatic Functions of Chymase on NAFLD/NASH
  3. Experimental Evidences on Chymase Inhibitors in NAFLD/NASH
  4. Role of Chymase on Metabolic Syndrome Complications
  5. Summary and Future Prospects

Round 2

Reviewer 1 Report

Takai and Jin significantly improved the quality of the review entitled "Chymase as A Possible Therapeutic Target for Amelioration of Non-Alcoholic Steatohepatitis".

I have no further comments

Reviewer 2 Report

The authors have made significant changes and now it can be accepted in its current form.

Reviewer 3 Report

OK